# Disparities in Lung Cancer: miRNA Isoform Characterization in Lung Adenocarcinoma

**DOI:** 10.3390/cancers14030773

**Published:** 2022-02-02

**Authors:** Rosario Distefano, Giovanni Nigita, Patricia Le, Giulia Romano, Mario Acunzo, Patrick Nana-Sinkam

**Affiliations:** 1Department of Cancer Biology and Genetics and Comprehensive Cancer Center, The Ohio State University, Columbus, OH 43210, USA; rosario.distefano@osumc.edu (R.D.); giovanni.nigita@osumc.edu (G.N.); 2Division of Pulmonary Diseases and Critical Care Medicine, Virginia Commonwealth University, Richmond, VA 23298, USA; patricia.le@vcuhealth.org (P.L.); giulia.romano@vcuhealth.org (G.R.)

**Keywords:** miRNA isoform, isomiR, miRNA editing, lung adenocarcinoma, race-disparities

## Abstract

**Simple Summary:**

Post-transcriptional modification events in miRNA molecules have revealed a more complex layer to cancer biology. Such modifications have created a novel path to developing and testing potential cancer biomarkers. Here, we concurrently profiled canonical and non-canonical miRNA molecules in White American (W) and Black or African American (B/AA) lung adenocarcinoma (LUAD) patients from The Cancer Genome Atlas (TCGA) cohort. We identified distinct potential post-transcriptional modifications in lung cancer tissues from W versus B/AA patients. Our results suggested the relevance of miRNA isoforms as potential biomarkers in lung cancer.

**Abstract:**

Despite the development of targeted therapeutics, immunotherapy, and strategies for early detection, lung cancer carries a high mortality. Further, significant racial disparities in outcomes exist for which the molecular drivers have yet to be fully elucidated. The growing field of Epitranscriptomics has introduced a new layer of complexity to the molecular pathogenesis of cancer. RNA modifications can occur in coding and non-coding RNAs, such as miRNAs, possibly altering their gene regulatory function. The potential role for such modifications as clinically informative biomarkers remains largely unknown. Here, we concurrently profiled canonical miRNAs, shifted isomiRs (templated and non-templated), and miRNAs with single-point modification events (RNA and DNA) in White American (W) and Black or African American (B/AA) lung adenocarcinoma (LUAD) patients. We found that while most deregulated miRNA isoforms were similar in W and B/AA LUAD tissues compared to normal adjacent tissues, there was a subgroup of isoforms with deregulation according to race. We specifically investigated an edited miRNA, miR-151a-3p with an A-to-I editing event at position 3, to determine how its altered expression may be associated with activation of divergent biological pathways between W and B/AA LUAD patients. Finally, we identified distinct race-specific miRNA isoforms that correlated with prognosis for both Ws and B/AAs. Our results suggested that concurrently profiling canonical and non-canonical miRNAs may have potential as a strategy for identifying additional distinct biological pathways and biomarkers in lung cancer.

## 1. Introduction

In the United States, lung cancer remains the leading cause of cancer death in males and females. In 2020, the American Cancer Society reported estimates of 235,000 new cases and 130,000 deaths from lung and bronchus cancers [1]. Annual incidence and mortality rates are further segregated based on race, ethnicity, and sex [2]. Lung cancer disproportionally affects Black or African American (B/AA) males, exemplified by higher incidence and mortality rates than in other reported racial and ethnic groups [2,3]. The intersectionality of many social, environmental, economic factors and, ultimately, potential genetic factors contribute to these racial disparities in lung cancer [2,4,5,6,7].

Unfortunately, B/AAs tend to present with late-stage disease when treatment is unlikely to be curative [8]. If diagnosed at an early stage, the survival rate of cancer patients is much higher (~70%) [8]. The National Lung Screening Trial (NLST) found that lung cancer screening (LCS) with low-dose CT (LDCT), as compared with chest radiography, was associated with a 20% reduction in mortality rate [9]. The findings from this study informed the recommendations of United States Preventative Services Task Force (USPSTF) for lung cancer screening (LCS) [10]. Under the 2013 USPSTF recommendations, individuals aged 55–80 years old with a minimum 30-pack/year smoking history are eligible for LCS (REF). However, numerous studies have since demonstrated that B/AAs are generally diagnosed with lung cancer at younger ages and fewer packs/year smoking histories [4,10,11]. In 2021, the USPSTF broadened its criteria for LCS to individuals aged 50–80 years old with a minimum of 20-pack/year smoking history [12]. One retrospective study compared the 2013 and 2021 USPSTF recommendations on a cohort of patients enrolled in the Inflammation, Health, Ancestry, and Lung Epidemiology (INHALE) study, and found that the racial disparities observed with using the 2013 criteria were absent when using the 2021 criteria [12]. Although smoking is implicated with a higher risk of developing lung cancer, it is clear that other factors contribute to the observed racial disparities in lung cancer incidence and mortality [6,8,11].

The emergence of large-scale sequencing technologies for profiling the human genome has allowed for comparative analyses across patient groups. Lower frequencies of all lung cancer driver mutations, along with higher tumor mutational burden, are found in B/AAs vs. whites (Ws) [13,14,15]. Whole-exome sequencing of B/AA non-small-cell lung cancer (NSCLC) patients without a known driver mutation revealed 47 novel and potentially damaging nonsynchronous mutations that were not found in The Cancer Genome Atlas (TCGA) [16]. This suggests that broadening current mutation profiling to encompass these novel driver genes may elucidate new targets for guiding clinical decisions [16].

Although several studies have examined the genomic landscape of lung cancers across racial groups, few studies so far have investigated for alterations in the non-coding transcriptome [17], such as miRNA isoforms [18]. Recent advances in high-throughput sequence technology have revealed an additional layer of complexity within miRNA molecules [19], paving the way for the field of “Epitranscriptomics” of ncRNAs in cancer [20]. The human genome predominantly encodes for non-coding RNAs (ncRNAs), such as microRNAs (miRNAs) [21,22]. MiRNAs are single-stranded RNAs, ~22 nucleotides in length, involved in post-transcriptional gene regulation primarily by an RNA interference mechanism [14]. During miRNA biogenesis, multiple cleavage events are catalyzed by RNases (e.g., Drosha, Dicer) to process primary miRNAs (pri-miRNA) into a mature miRNA duplex [23]. High-throughput sequencing has revealed that various miRNA isoforms (isomiRs) may be derived from the same precursor miRNA (pre-miRNA) [23]. These isoforms appear from alternative processing by Drosha and Dicer, as well as post-transcriptional modifications or “RNA editing” events, which include non-templated nucleotide additions (NTA) (e.g., adenylation, uridylation) and nucleotide substitutions (e.g., cytosine-to-uridine/C-to-U, adenosine-to-inosine/A-to-I) [23,24,25,26]. Moreover, when these edits occur within the functional 5′ seed region of miRNAs (from nucleotides 2–7), miRNA-mRNA target interactions are altered [27,28], thereby creating new patterns of targeting that have been shown to affect downstream biology [23,24,26,28,29,30,31]

The spatiotemporal nature of RNA editing can provide insight into the dynamic nature of human cancers. [29,32]. Generally, human cancers display hypo-editing of miRNAs, consequently upregulating their targets [29,31]. Expression of miRNA isoforms in triple-negative breast cancer (TNBC) tissue samples from a TCGA dataset between B/AA and W patients demonstrated differences in isoforms from loci of known oncogenic miRNAs (oncomiRs) (e.g., miR-200c, miR-21, miR-17/92 family, miR-183/96/182 family) [18]. This differential expression of isomiRs may alter the dynamics of metastatic pathways (e.g., P38, Wnt/β-catenin), contributing to the aggressive phenotype of TNBC [18].

The conventional approach to studying miRNA expression is to cluster miRNA isoforms within a shifting range of a few nucleotides (miRNA-arms). Unlike this approach, we concurrently profiled canonical miRNAs, shifted isomiRs (templated and non-templated), and miRNAs with single-point modification events (e.g., A-to-I RNA editing, SNPs, and nucleotide insertions) in lung adenocarcinoma (LUAD) samples from TCGA data sets. In particular, given the racial disparities prevalent in lung cancer, we assessed the differential expression of canonical and non-canonical miRNAs between B/AA and W from the LUAD-TCGA cohort. Among the deregulated miRNA isoforms, we elected to investigate the edited form of miR-151a-3p, with an A-to-I editing event in the seed region at position three, in W versus B/AA LUAD tissues. Here, we demonstrated, based on pathway enrichment analysis, that edited miR-151a-3p has a potential differential biological effect in B/AA versus W LUAD lung cancers.

Moreover, we identified additional miRNA isoform expression patterns that significantly and distinctly stratify patients in overall survival (OS) and relapse-free survival (RFS) in B/AA versus W patients with lung adenocarcinoma. These results suggested that racial intragroup and intergroup differences in miRNA isoform expression may serve as additional classifiers in the development of clinically meaningful biomarkers.

## 2. Results

### 2.1. Systematic miRNA Isoform Characterization in Black or African and White American Lung Adenocarcinoma Patients

To investigate expression patterns of miRNA isoforms in both B/AA and W lung adenocarcinoma patients, we analyzed small-RNA sequencing (sRNA-seq) sample data available from the LUAD-TCGA cohort. In particular, we examined data from 485 tissues, precisely 58 B/AA (52 primary solid tumors and six adjacent normal tissues) and 427 W (387 primary solid tumors and 40 adjacent normal tissues) samples. The cohort characteristics are summarized in Appendix A.

In order to concurrently detect and quantify canonical miRNAs, miRNAs with single nucleotide variant (SNV) events (e.g., Single Nucleotide Polymorphisms (SNPs), A-to-I RNA editing, and nucleotide insertions), and templated/non-templated shifted isomiRs (see Figure 1A), we performed an in-house workflow (see Appendix A and Materials and Methods section) which implements the miRge 2.0 pipeline [33]. We also created a specific nomenclature for the detected miRNA isoforms based on the miRGff3 [34] format (see Appendix A and Materials and Methods for more details). SNVs occurring in miRNA sequences were further annotated with A-to-I RNA editing sites (MiREDiBase database [35]), somatic mutations (COSMIC database [36], and TCGA and TARGET cohorts), and known SNPs (dbSNPs) [37]. Furthermore, we employed a filtering phase, discarding miRNA isoforms with a not-annotated SNV in the first or last two nucleotides (see Materials and Methods for more details). Finally, we identified a total of 5582 expressed miRNA isoforms (see Figure 1B, Appendix A and Materials and Methods section for more details) in the LUAD-TCGA cohort (3785 in W normal tissues, 3832 in W tumor tissues, 4137 in B/AA normal tissues, and 3881 in B/AA tumor tissues). Of note, one of the expressed molecules was a novel miRNA, labeled as *miR-n47-3p* (Genomic coordinates: chr19, 13836509-13836528, negative strand).

In all four sample groups, the miRNA isoforms with a shift in the 3’end with no SNVs were the most representative ones, which were >3.5 times more than the canonical miRNA molecules (Figure 1B). Interestingly, considering the expression of each of 16 miRNA isoform categories, the class of the miRNA isoforms with a 3’end shifting with no SNVs had an expression comparable to the expression of canonical molecules (see Appendix A), rendering these isoforms attractive from a functional standpoint.

Furthermore, intersecting the expressed miRNA isoforms identified in each group sample, we found >2.2 K miRNA isoforms in common (Figure 1C). Moreover, we identified some miRNA isoforms which were specifically expressed in each group: 145 in W(N), 120 in W(T), 459 in B/AA(N), and 230 in B/AA(T). Interestingly, our results showed that B/AA samples have more miRNA isoforms distinctly expressed than W samples, highlighting a disparity between the two races, which, interestingly, was not evident if we take into consideration only canonical molecules (see Appendix A).

### 2.2. miRNA Isoform Dysregulation in Lung Adenocarcinoma Race Disparities

To identify potential dysregulated miRNA isoforms in lung adenocarcinoma across races, we performed a multivariate analysis comparing primary solid tumor (T) and normal adjacent (N) tissues from both W and B/AA LUAD patient samples. The raw read counts of the expressed miRNA isoforms (⌊RPM geometric mean⌋ > 1 Reads Per Million across the four sets of samples) were first normalized, and then we performed a multivariate analysis (see Materials and Methods for more details), considering four distinct sample groups: (I) W Normal, (II) W Tumor, (III) B/AA Normal, and (IV) B/AA Tumor tissue samples.

We obtained a total of 2935 deregulated miRNA isoforms (FDR < 0.05), which we subdivided into six different trends based on the trend in the four groups mentioned above (see Figure 2 and Appendix A): (I) monotonic increasing, meaning that the average (specifically, the geometric mean) miRNA isoforms increase along four tissue types (from W Normal to B/AA Tumor tissue samples); (II) monotonic decreasing, indicating that the average of miRNA isoforms decrease along the four tissue types; (III) upregulated in both races, that are the miRNA isoforms upregulated in both W and B/AA patients, and are not monotonic increasing; (IV) downregulated in both races, namely the miRNA isoforms downregulated in both W and B/AA patients, and they are not monotonic decreasing; (V) Upreg. in W, Downreg. in B/AA, considering the miRNA isoforms upregulated in W patients and, at the same time, downregulated in B/AA patients; and (VI) Downreg. in W, Upreg. in B/AA, representing the miRNA isoforms downregulated in W patients and, at the same time, upregulated in B/AA patients. Our results showed that upregulated and downregulated miRNA molecules in both races were the most representative trends (>94%). Nevertheless, a few miRNA molecules (~6%, primarily non-canonical ones) fell into one of the four remaining trends (monotonic or opposite trends), showing specific differences between W and B/AA.

### 2.3. Dysregulation of Edited miR-151a-3p in Black or African and White American Lung Adenocarcinoma Patients: A Case of Study

To study the potential impact of deregulated non-canonical miRNAs on downstream targets, we examined the edited form of miR-151a-3p with an A-to-I editing event in position 3. MiR-151-3p was highly expressed in all four distinct tissue types, compared with the other miRNA isoforms (W(N)~89th percentile, W(T)~87th perc., B/AA(N)~90th perc., and B/AA(T)~89th perc.). Moreover, as shown in Appendix A, it was down-regulated in both W and B/AA tumor samples, although more marked in B/AA tumor samples (based on the geometric mean of each group). We sought to determine if the edited miRNA had a differential impact, even though the trend in expression was similar in both races. Interestingly, downregulation of edited miR-151a-3p correlated with the downregulation of the adenosine deaminase RNA-specific B1 (ADARB1 or ADAR2) gene, but not with the ADAR (ADAR1) gene (see Appendix A), which is consistent with previous observations that ADAR2 deregulation could drive such editing events in miR-151a-3p [31,38,39]. In addition, the edited miRNA molecule has an A-to-I editing event in position 3 (in the miRNA seed region—MSR), which could result in a downstream targetome distinct from that of the canonical miR-151a-3p, as is the case in other miRNAs that we and others have demonstrated in the past [27,28,40].

In order to uncover the potential downstream impact of the edited miR-151a-3p in both W and B/AA patient samples, we first subdivided the tumor samples (for both races) into two distinct groups based on the low (first quartile, Q1) and high (third quartile, Q3) expression of the miRNA molecule (see Figure 3A). We then employed a differential expression analysis of genes, comparing Q1 vs. Q3 groups (see Figure 3B and Appendix A). As we were interested in the impact on gene expression of the miRNA molecule, we considered all those significant D.E. genes (*p*-value < 0.01 and |linear fold change| > 1.5, see Materials and Methods section for more details) which potentially are direct or indirect targets. Subsequently, we performed a pathway enrichment analysis using the Ingenuity^®^ Pathway Analysis (IPA) software (see Figure 3C and Appendix A) based on the D.E. genes.

In summary, we demonstrated that the expression of edited miR-151a-3p is associated with different canonical pathways comparing the two patient groups (W and B/AA LUAD patients). Indeed, only a few canonical pathways were shared between the two population groups (see Figure 3C).

### 2.4. Prognostic miRNA Isoform Signature in Lung Adenocarcinoma Race Disparities

Considering the significant number of expressed miRNA isoforms identified through the concurrent proofing of canonical and non-canonical miRNAs (see Figure 1B), we decided to investigate the potential of such miRNA isoforms as clinical biomarkers. To accomplish this goal, we designed a two-stage workflow (see Appendix A and Materials and Methods) to identify prognostic signatures for Overall (OS) and Relapse-Free (RFS) survival.

As shown in Figure 4A, we identified two different signatures for OS, with 13 miRNA isoforms (including three canonical miRNAs) for W LUAD patients and five miRNA isoforms (including one canonical miRNA) for B/AA LUAD patients (see Appendix A). We also tested these prognostic signatures in the other race group, finding that the 13-miRNA-isoform signature identified in W LUAD patients carried similar accuracy for B/AA LUAD samples (the AUC is 0.65 for both race groups); in contrast, the 5-miRNA-isoform signature identified in B/AA LUAD patient appeared to be more accurate for B/AAs (AUC = 0.78) compared to Ws (AUC = 0.56).

Similar to the OS analyses, we identified two distinct miRNA isoform signatures for RFS for W and B/AA LUAD patients (see Figure 4B). In particular, these were an RFS signature harboring ten miRNA isoforms (including a canonical miRNA) for W LUAD patients and a signature for B/AA LUAD (see Appendix A). Testing the RFS signatures in the other group, we determined that the ten-miRNA-isoform signature for W LUAD patients (AUC = 0.79) had similar accuracy for B/AA LUAD patients (AUC = 0.74); however, the miRNA isoform signature for B/AA LUAD patients was not expressed within W LUAD patient tissues and, thus, was not testable as a potential biomarker.

In conclusion, through the analysis of both canonical and non-canonical miRNA molecules, we identified distinct patterns of expression of isoforms between B/AA and W LUAD patients that harbor clinical significance. These results suggested that investigation of miRNA isoforms may serve as an additional avenue for identifying clinically informative biomarkers and, more importantly, elucidating novel cancer biology.

## 3. Discussion

Disparities in lung cancer diagnosis and treatment have contributed to widening the gap in care and outcomes [2]. In particular, Black or African American (B/AA) males are disproportionately burdened by this disease, as evidenced by higher incidence and mortality rates [2,3]. Under the 2013 USPSTF recommendations, B/AAs were less likely to meet the criteria for lung cancer screening (LCS) by low-dose CT (LDCT) [4,5,6,17]. B/AAs are generally diagnosed at a later stage of disease, at a younger age, and with a lower pack-per-year smoking history when compared with White Americans (W) [4,5,6,17]. In 2021, the USPSTF broadened its criteria to include individuals aged 50–80 years old with a minimum of 20-pack/year smoking histories; the implementation of the 2021 USPSTF criteria in retrospective studies resulted in previous racial disparities in LCS being no longer observed [12]. Although cigarette smoking is a major risk factor for the development of lung cancer, it is clear that other factors contribute to disparities associated with this disease Therefore, we have proposed to evaluate the role of the epitranscriptome in lung cancer disparities.

The advent of high-throughput sequencing platforms has driven the discovery of non-coding RNAs (ncRNAs), such as microRNAs (miRNAs) [21,22], as both drivers of cancer biology and clinical biomarkers. The dysregulation of miRNA expression has been well-documented in all human cancers. Previous studies have mainly focused on evaluating canonical miRNAs, which represent excellent diagnostic cancer biomarkers if they are considered as single molecules instead of clusters. One limitation of this approach is that only a few hundred canonical miRNAs are expressed. In light of this, we concurrently profiled both canonical and non-canonical miRNA molecules to significantly expand the number of potential cancer biomarkers from several hundred to thousands. The inclusion of non-canonical miRNAs could significantly improve the resolution of potential bio-classifiers. This will be particularly valuable for elucidating subtle molecular differences that may contribute to lung cancer disparities.

Here, we profiled canonical miRNAs, shifted isomiRs (templated and non-templated), and miRNAs with single nucleotide variant (SNV) events (e.g., single nucleotide polymorphisms (SNPs), A-to-I RNA editing, nucleotide insertions) in lung adenocarcinoma (LUAD) samples available from the TCGA (see Figure 1). We conducted differential expression analysis in order to compare the levels of miRNA isoforms between B/AA and W LUAD patients (see Figure 2). We elected to study the function of edited miR-151a-3p, which undergoes an A-to-I editing site in the seed region at position 3. This approach was based on similar studies by us and others who have previously investigated the functional consequences of miRNAs that possess an editing site within the seed region [27,28,40]. Our results here demonstrated that the differential expression of a single miRNA isoform (edited miR-151a-3p) in B/AAs and Ws may be associated with a distinct set of pathways (see Figure 3). However, not all sequence shifts observed between miRNA isoforms will alter the original function. Instead, these isomiRs can serve as candidate classifiers. Therefore, we sought to identify miRNA isoforms that classify patients in terms of overall survival (OS) and relapse-free survival (RFS). Intriguingly, we have found two different prognostic signatures of OS and RFS that could distinguish B/AAs but not Ws LUAD patients (see Figure 4), illustrating the importance of considering racial differences in miRNA isoform expression when developing clinical classifiers.

Altogether, the results obtained from these studies demonstrate the potential of profiling a wider miRNAome. We conclude that concurrently profiling canonical and non-canonical miRNAs may provide additional insights into cancer biology and clinically useful biomarkers. We recognize the limitations of the current study, including, most importantly, the relatively low number of cases of B/AA lung cancer within the TCGA database, the possibility that observed differences in isoforms may be driven by factors other than race, and the need for independent validation of any clinical classifier. Nevertheless, the encouraging findings from our study suggested that a more comprehensive approach to miRNA profiling that incorporates both canonical and non-canonical forms may prove valuable in identifying additional molecular mechanisms of lung cancer.

## 4. Materials and Methods

### 4.1. Data Acquisition and Preparation

sRNA-seq sequencing BAM files of 485 lung adenocarcinoma tissue samples (TCGA-LUAD project, with 439 primary solid tumors and 46 adjacent normal tissues) were downloaded via the GDC data portal (https://gdc-portal.nci.nih.gov, accessed on: 5 February 2019). Before the detection analysis, we converted BAM files into FASTQ ones (which contain raw, unaligned reads with quality scores) by using the bamtofastq function from BEDTools (v2.25.0) package [41]. Finally, all the raw sequence data were quality-filtered using the ConDeTri tool (v2.3) [42], with the following parameters: −pb = fq −lq = 20 −hq = 30 −minlen = 15 −sc = 33.

We also downloaded raw read count and FPKM values (where FPKM means Fragments Per Kilobase Million) of genes detected from RNA-seq samples (matched with the sRNA-seq samples described above) from the TCGA-LUAD project via the GDC data portal.

### 4.2. MiRNA Isoforms Mapping, Annotation, and Filtering

Quality-filtered sequences were aligned to the human genome (hg38) and, subsequently, annotated via an in-house designed workflow (Appendix A). The mapping and annotation phases leveraged on miRge 2.0 [33] tool, a pipeline for canonical miRNAs/miRNA isoforms annotation based on miRGeneDB 2.0 [43] and the miRBase (v22) [44]. The miRge 2.0 pipeline was executed using the default parameters, performing the *prediction* module to discover potential novel miRNA molecules. As illustrated in Figure 1A, annotated miRNA molecules include canonical miRNAs, shifted miRNAs, and miRNA with single nucleotide variants (SNVs), such as A-to-I RNA Editing sites and Single Nucleotide Polymorphisms (SNPs), and somatic mutations. After the annotation phase, the annotated miRNA molecules were enriched with additional information, such as known SNPs (dbSNP v154 [37]), somatic mutations (COSMIC database v92 [36], and TCGA and TARGET cohorts), and A-to-I RNA editing sites from the MiREDiBase (v1.0) database [35]. Moreover, to eliminate potential imperfection in the linker ligation during the cDNA library construction or sequencing errors [45,46,47], we discarded all those miRNA isoforms with at least a not-annotated SNV in the first or last two nucleotides.

The miRNA isoform nomenclature uses the information from the mirGFF3 format [34], which was extended with other features. More details about the miRNA isoform nomenclature are illustrated in Appendix A.

Finally, data were collected into tab-separated text files considering those miRNA isoforms with an expression of ⌊RPM geometric mean⌋ > 1 (where RPM means Reads Per Million) for the downstream analyses.

### 4.3. miRNA Isoform Quantification and Multivariate Analysis in LUAD Race Disparities

Raw read counts of the expressed miRNA isoforms were normalized via the trimmed mean of M-values (TMM) [48] method by employing the calcNormFactors function available in the edgeR (v3.32.1) [49,50] R (v3.4.4) package from *BioConductor* (v3.6) [51].

In the determination of the expressed miRNA isoforms for each group (W (N), W (T), B/AA (N), and B/AA (T)), the read counts of the miRNA isoforms were normalized with the TMM method separately for each group (see Figure 1B,C).

Concerning the multivariate analysis, using the normalized data (normalizing all the four sample groups together), we performed a Kruskal-Wallis test comparing the four LUAD sample groups (see Figure 2). Finally, each *p*-value was adjusted by Benjamini-Hochberg’s correction method [52].

### 4.4. Differential Gene Expression and Functional Enrichment Analyses

Firstly, for both W and B/AA LUAD patients, we grouped samples according to the first (Q1) and third (Q3) quartile of edited miR-151a-3p to identify deregulated genes related to the expression edited miRNA molecule (Q1 vs. Q3) considered in our study. Before the normalization step, low expressed genes were filtered out, retaining those genes with a minimum expression of ⌊FPKM geometric mean⌋ > 1. Subsequently, the set of expression-filtered genes was used to extract the corresponding raw read counts and then normalized via TMM [48]. Finally, for both W and B/AA LUAD patients, the differential expression analyses were performed by using edgeR. All those dysregulated expressed (D.E.) genes with *p*-value < 0.01 and |linear fold change| > 1.5 were considered for the functional enrichment analyses (see Appendix A). Later, a pathway enrichment analysis was performed employing the Ingenuity^®^ Pathway Analysis (IPA^®^) software (v01-16) based on the D.E. genes. All those pathways with a *p*-value < 0.01 were retained (see Appendix A).

### 4.5. Prognostic Signature Identification

The right-censored data were considered for the patient’s survival time for overall (OS) and relapse-free (RFS) survival analyses. The prognostic miRNA isoform signature workflow consisted of two main stages, as shown in Appendix A.

In the first stage, the patients’ miRNA isoforms expression data were assembled, starting by whether the sample group (either W or B/AA patients) has at least ten patients per event type: “dead” and “alive” for the OS and relapse or not-relapse for RFS. Subsequently, the raw read counts were taken for the selected patients and then normalized with the method described in Section 4.4. Finally, a correlation matrix was calculated out of the normalized data.

The second stage started with removing highly correlated miRNA isoforms, retaining those miRNA molecules with a correlation lower than a specific threshold, ranging from 0.6 to 0.8 (with incremental steps of 0.01). At this point, a first step generates a univariate Cox proportional hazard regression model, assessing the relationship between miRNA isoforms expression and patients’ survival. Each *Cox p*-value was adjusted using Benjamini-Hochberg’s method. Subsequently, a multivariate Cox proportional hazard regression model was performed based on the most significant miRNA isoforms (*Cox FDR* < 0.05) coming from the univariate model. Both univariate and multivariate Cox regression models leverage the CoxPHFitter function from the lifelines (v0.25.4) Python package. The list of miRNA isoforms present in this stage was narrowed down by keeping those significant miRNA molecules (with a *Cox p*-value < 0.05 according to the multivariate model) and then applying a feature selection strategy. The selection strategy consisted of the Recursive Feature Elimination method from the scikit-learn (v0.22.1) Python package, using a Logistic Regression Model used as the estimator. Successively, for each patient, a Risk Score was computed by linearly combining the expression value and the regression coefficient (univariate model) associated with the reduced set of miRNA isoforms (selection strategy), as defined below:$$Riskscore=\overset{n}{\underset{i=1}{\sum }}exp_{i}\cdot \beta_{i}$$

In which *n* is the number of reduced miRNA isoforms, *exp* indicates the expression, and *β* represents the regression coefficient (from the univariate Cox model). Patients were then separated into high- and low-risk groups using the median of Risk Scores as a cutoff. Finally, in last step the Kaplan-Meier plot was generated, calculating the *p*-value (Log Rank Test) and a prognostic signature accuracy (area under the curve—AUC) based on the two risk groups. The Kaplan-Meier plots were generated using the *survfit* and *ggsurvplot* functions from the survival (v3.2-3) R package. The prognostic signature with the highest AUC was selected at the end of the workflow.

### 4.6. Statistical Analyses and Graphs

Venn diagrams in Figure 1B, Figure 3C, and Appendix A were created by using the InteractiVenn webtool [53]. The heatmaps present in Figure 1 and Figure 2 were generated using the matplotlib and seaborn Python libraries. Regarding the heatmaps, a miRNA molecule could be counted more than one time if it had more types of SNVs (e.g., known DNA variant, A-to-I RNA editing event, or unknown SNV).

## 5. Conclusions

In summary, we profiled miRNA isoforms in 485 lung adenocarcinoma (LUAD) samples, specifically from 427 White American and 58 Black or African American patient donors, to investigate the role of miRNA isoform in lung cancer race disparities. Unlike conventional methods, we concurrently profiled canonical and non-canonical miRNA isoforms, discovering more miRNA isoforms expressed explicitly in B/AA samples compared with Ws. Most of the deregulated miRNA isoforms deregulated LUAD samples conserved the same trend (up-or down-regulation) between the two race groups, and just a few miRNA molecules had an exclusive race-related trend. We also analyzed an expressed edited miRNA as a case study, showing how its expression related to different pathways between W and B/AA LUAD patients.

Moreover, based on miRNA isoform molecules, we discovered two distinctive overall survival signatures, one W and one for B/AA LUAD patients, and two different relapse-free survival signatures, one for both race-group patients. In particular, the prognostic signatures were unique to B/AA LUAD patients, suggesting that these miRNA isoforms require further examination and validation as potential race-specific prognostic biomarkers in lung cancer.

## Figures and Tables

**Figure 1 cancers-14-00773-f001:**
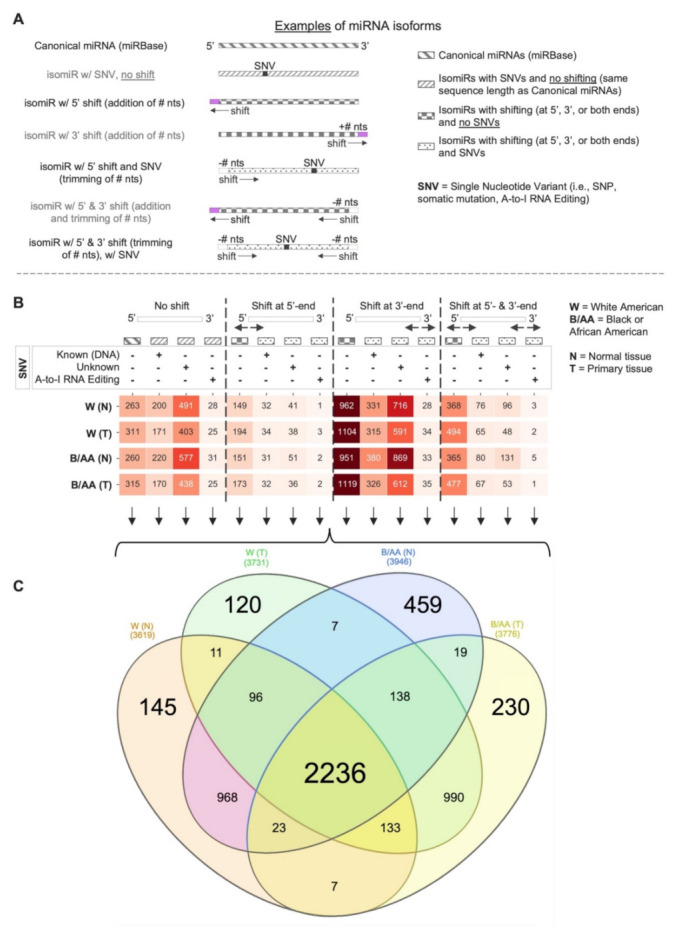
Canonical and non-canonical miRNAome profiling in lung adenocarcinoma. (**A**) Examples of miRNA isoforms detected from sRNA-seq data. (**B**) Distribution of expressed miRNA isoforms in primary solid tumor (T) and normal (N) tissues from both W and B/AA LUAD patient samples. The heatmap shows the number of expressed miRNA isoforms (canonical and non-canonical ones) for each of the four miRNA isoform categories (no shift, shift at 5′-end, shift at 3′-end, and shift at 5′- and 3′-end). (**C**) Venn diagram indicates the commonly expressed miRNA isoforms between the four group samples: W (N), W (T), B/AA (N), and B/AA (T).

**Figure 2 cancers-14-00773-f002:**
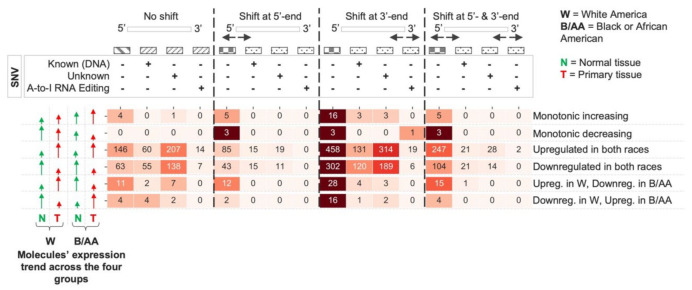
Deregulated miRNA isoforms in both W and B/AA lung adenocarcinoma patient samples. The heatmap shows the number of deregulated miRNA isoforms (canonical and non-canonical ones) for each of the four miRNA isoform categories (no shift, shift at 5′-end, shift at 3′-end, and shift at 5′- and 3′-end). Each trend is defined based on the geometric mean of each group.

**Figure 3 cancers-14-00773-f003:**
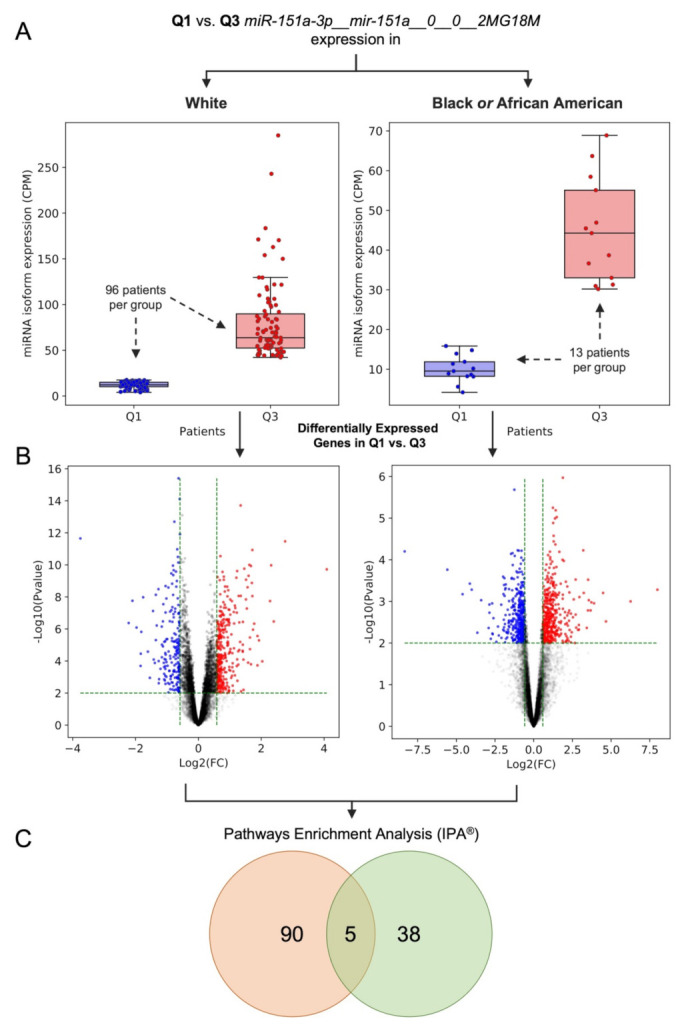
Different gene and pathway dysregulation related to A-to-I (in position 3) edited miR-151a-3p expression in LUAD race disparities. (**A**) Box plots indicate the expression of edited miR-151a-3p in both W and B/AA tumor samples with low (first quartile, Q1) and high (third quartile, Q3) expression of the miRNA molecule. (**B**) Scatter plots show the D.E. genes (*p*-value < 0.01 and |linear fold change| > 1.5) related to the expression edited miRNA (Q1 vs. Q3). (**C**) Venn diagram indicates the common significant enriched pathways (based on the D.E. genes) between W and B/AA patient samples.

**Figure 4 cancers-14-00773-f004:**
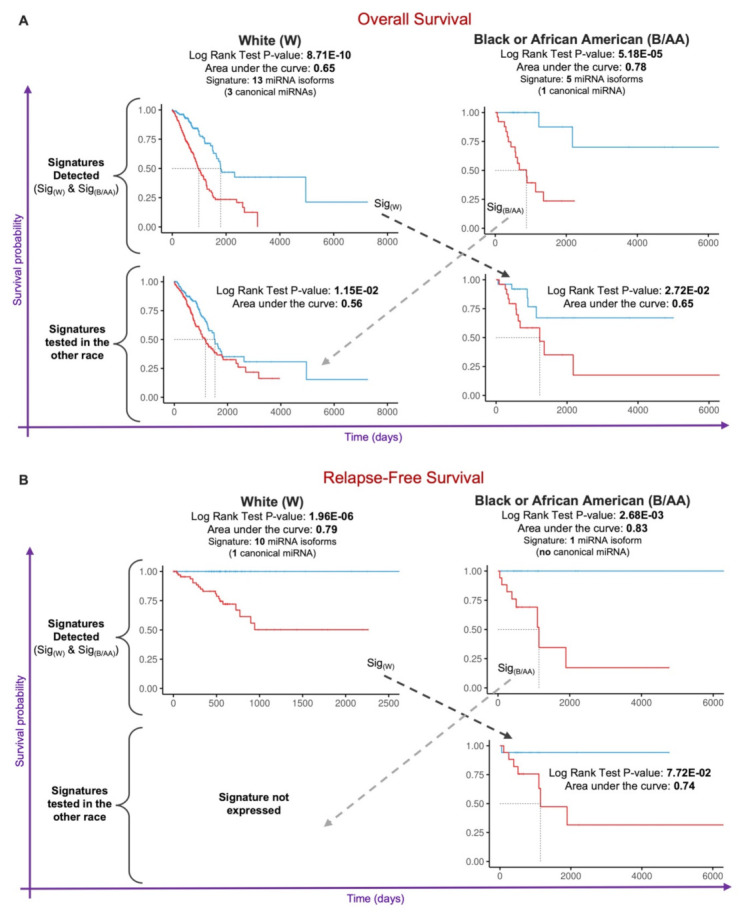
miRNA isoform prognostic signatures in lung adenocarcinoma across race, supplied with significance (Log Rank test *p*-value) and the area under the curve (AUC) score. (**A**) miRNA isoforms signatures for overall survival in W and B/AA patient samples. (**B**) miRNA isoform signatures for relapse-free survival in W and B/AA patient samples. The bottom part of each panel shows the results of the signatures test in the other race.

## Data Availability

All data are available in Appendix A. Raw sequence data were retrieved via the GDC data portal after obtaining authorization from the data access committee (DBGap Project ID: 11332).

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
