# Peer review of "Disparities in Lung Cancer: miRNA Isoform Characterization in Lung Adenocarcinoma"

_cancers, 2022, doi:10.3390/cancers14030773_

Round 1

Reviewer 1 Report

This paper presents exciting progress in making medicine a more equitable experience across racial identities. It is written as medicine is becoming more personalized to provide patients with the best care possible for their unique genetic and environmental circumstances and meets the moment for addressing disparities in how lung cancer has been treated for decades. MicroRNA research is relatively new and its use as an identifier of tissue injury or illness is showing quite a bit of promise, so I see why this particular avenue of research was attractive and practical for the authors.

Abstract/Simple Summary: I see no major issues with the summary or abstract. The summary lays out the research in as basic terms as possible, while the abstract outlines the key biological and social problems that inspired the research. The information presented in each was appropriate and gave the readers the “50,000-foot view” necessary to understand the basic focus of the paper. Most importantly, the authors incorporated the use of miRNA biomarkers in lung adenocarcinoma, which made this research extremely relevant.

Introduction: The introduction opens with relevant statistics of why lung cancer is a problem in the United States. The main issue I have with the early introduction is that they reference the “smoker’s paradox” with a very basic definition, but some further information or even an example would be beneficial here. I would also be interested in some contextual information regarding when/how the criteria for cancer screening was developed and when it was last updated so that readers have that frame of reference. This team does give appropriate attention to the different intersectional aspects of health that cause the detection and treatment of lung cancer to have disparities across races. The introduction to miRNA origin and processing is fantastic, especially since they did it in such a small space.

Results

2.1: Systematic miRNA Isoform Characterization in Black or African American and White American Lung Adenocarcinoma Patients: The method of using small-RNA sequencing is quite sound. It would be beneficial for the authors to lay out why they chose the number of patients that they did from each patient cohort since there is a large difference in the number of data points generated by white American patients vs. Black or African American. There is no issue with the heatmap presented in 1B and it is clearly organized. Things get a little harder to follow with figure 1C. The Venn diagram is a favorite tool of bioinformaticians to display common gene (or miRNA) expression between experimental groups, and it is often the best option for display. This diagram in particular is difficult to follow because there are so many similar colors and the most important numbers do not stand out. Consider using highlighting or bolding to make the shared expressed miRNAs that are most relevant to this result stand out a little more.

2.2: miRNA Isoform Dysregulation in Lung Adenocarcinoma Race disparities: Figure 2 represented in this sub-section was well planned and the color coding makes it pretty easy to follow. I see no major issues in this section

2.3: Dysregulation of edited miR-151a-3p in Black or African and White American Lung Adenocarcinoma Patients: Check line 201-is it supposed to say “conical” or “canonical”. In Figure 3A, the boxplots are neat and easily interpreted, however it may be helpful to revisit the discrepancy between white and Black or African American patients (in terms of numbers of data points) in this sub-section.

2.4: Prognostic miRNA Isoform Signature in Lung Adenocarcinoma Race Disparities: This section packs the most punch in my opinion. These prognostic isoform signatures have the potential to be an absolute game changer in terms of lung adenocarcinoma diagnosis and treatment, particularly for Black or African Americans.

Discussion: In the opening paragraph, the authors address the socioeconomic components unrelated to cigarette smoking that put Black or African American males at a higher incidence of lung adenocarcinoma but do not address these components. It would be helpful to the reader if some of these components related to SES were addressed in the introduction, then revisited in the discussion. Additionally, in the opening paragraph the authors reference a more comprehensive approach to assessing lung cancer risk and screening for lung tumors in traditionally underserved populations, but they don’t make recommendations about how to do this.

Figure S5- I think that the color purple represents the normal tissue type for both white and Black or African American patients while the pink color represents tumorous tissue across races. A simple key would add some clarity.

Author Response

Reviewer 1

  • Comments and Suggestions for Authors

This paper presents exciting progress in making medicine a more equitable experience across racial identities. It is written as medicine is becoming more personalized to provide patients with the best care possible for their unique genetic and environmental circumstances and meets the moment for addressing disparities in how lung cancer has been treated for decades. MicroRNA research is relatively new and its use as an identifier of tissue injury or illness is showing quite a bit of promise, so I see why this particular avenue of research was attractive and practical for the authors.

Abstract/Simple Summary: I see no major issues with the summary or abstract. The summary lays out the research in as basic terms as possible, while the abstract outlines the key biological and social problems that inspired the research. The information presented in each was appropriate and gave the readers the “50,000-foot view” necessary to understand the basic focus of the paper. Most importantly, the authors incorporated the use of miRNA biomarkers in lung adenocarcinoma, which made this research extremely relevant.

Response: We appreciate the time and effort that the Reviewer dedicated to providing feedback on our manuscript. Please see attached the revised version of our manuscript. To get our changes in the text quickly, we have labeled the modifications in Red color.

  • Introduction: The introduction opens with relevant statistics of why lung cancer is a problem in the United States. The main issue I have with the early introduction is that they reference the “smoker’s paradox” with a very basic definition, but some further information or even an example would be beneficial here. I would also be interested in some contextual information regarding when/how the criteria for cancer screening was developed and when it was last updated so that readers have that frame of reference. This team does give appropriate attention to the different intersectional aspects of health that cause the detection and treatment of lung cancer to have disparities across races. The introduction to miRNA origin and processing is fantastic, especially since they did it in such a small space.

Response: We thank the Reviewer for the kind words and suggestions. The “Smoker’s Paradox” is a general observation where certain racial groups have a disproportionate burden of disease despite having fewer pack/year smoking histories. This paradox is well-reported for Black/African American patients with lung cancer. We have elected to remove the term “Smoker’s Paradox” while retaining the overall message of “Smoker’s Paradox.” Per the Reviewer’s suggestion, we have expanded the portion on lung cancer screening to include information on how the recommendations by the USPSTF were developed and modified in recent years.

  • 1: Systematic miRNA Isoform Characterization in Black or African American and White American Lung Adenocarcinoma Patients: The method of using small-RNA sequencing is quite sound. It would be beneficial for the authors to lay out why they chose the number of patients that they did from each patient cohort since there is a large difference in the number of data points generated by white American patients vs. Black or African American. There is no issue with the heatmap presented in 1B and it is clearly organized. Things get a little harder to follow with figure 1C. The Venn diagram is a favorite tool of bioinformaticians to display common gene (or miRNA) expression between experimental groups, and it is often the best option for display. This diagram in particular is difficult to follow because there are so many similar colors and the most important numbers do not stand out. Consider using highlighting or bolding to make the shared expressed miRNAs that are most relevant to this result stand out a little more.

Response: We appreciated the Reviewer's comment. Following the Reviewer’s suggestion, we edited the Venn diagram to change the colors to mark the intersections between the different sets better. We have also increased the font size of some numbers we have mentioned in the text. In particular, the number of common miRNA isoforms and the number of those detected only in a specific cohort.

Regarding the number of samples, we agree with the Reviewer. There is a large difference between B/AA and W patients samples. Unfortunately, these are the data available in the TCGA-LUAD cohort, indicating a certain disparity even in the data collected. The reasons behind this discrepancy could be different, such as either social or economic aspects. However, we have commented this limitation at the end of the Discussion section, saying:

We recognize the limitations of the current study, including, most importantly, the relatively low number of cases of B/AA lung cancer within the TCGA database, the possibility that observed differences in isoforms may be driven by factors other than race, and the need for independent validation of any clinical classifier.

Furthemore, we have added the word “available” in the “2.1. Systematic miRNA isoform characterization in Black or African and White American lung adenocarcinoma patients” section (line 146) to indicate these were the only samples available in the TCGA-LUAD cohort.

  • 2: miRNA Isoform Dysregulation in Lung Adenocarcinoma Race disparities: Figure 2 represented in this sub-section was well planned and the color coding makes it pretty easy to follow. I see no major issues in this section

Response: Thank you for your valuable comment.

  • 3: Dysregulation of edited miR-151a-3p in Black or African and White American Lung Adenocarcinoma Patients: Check line 201-is it supposed to say “conical” or “canonical”. In Figure 3A, the boxplots are neat and easily interpreted, however it may be helpful to revisit the discrepancy between white and Black or African American patients (in terms of numbers of data points) in this sub-section.

Response: Thank you for your suggestions. We made the change suggested by the Reviewer in the revised version of the manuscript. Regarding the number of B/AA and W sample data, as we replied above, the discrepancy between B/AA and W patients is due to the available samples data in the TCGA-LUAD cohort.

  • 4: Prognostic miRNA Isoform Signature in Lung Adenocarcinoma Race Disparities: This section packs the most punch in my opinion. These prognostic isoform signatures have the potential to be an absolute game changer in terms of lung adenocarcinoma diagnosis and treatment, particularly for Black or African Americans.

Response: Thank you for your valuable comment.

  • Discussion: In the opening paragraph, the authors address the socioeconomic components unrelated to cigarette smoking that put Black or African American males at a higher incidence of lung adenocarcinoma but do not address these components. It would be helpful to the reader if some of these components related to SES were addressed in the introduction, then revisited in the discussion. Additionally, in the opening paragraph the authors reference a more comprehensive approach to assessing lung cancer risk and screening for lung tumors in traditionally underserved populations, but they don’t make recommendations about how to do this.

Response: We wanted to illustrate that racial disparities in lung cancer is a multi-faceted and complex issue that requires a holistic approach where multiple factors are considered. The scope of our research is primarily focused on understanding the role of the epitranscriptome in lung cancer disparities. We agree with the Reviewer’s comments that these additional factors were not well-discussed in the manuscript. As such, we decided to eliminate the reference to a “more comprehensive approach” and have rewritten this portion so that it better represents the scope of our research.

  • Figure S5- I think that the color purple represents the normal tissue type for both white and Black or African American patients while the pink color represents tumorous tissue across races. A simple key would add some clarity.

Response: We appreciated the Reviewer’s comment, and we edited the legend of Figure S5 for a better understanding.

Reviewer 2 Report

The work presented by Distefano et al. relates specific miRNA isoforms expression patterns that significantly stratify patients in overall survival and relapse-free survival in B / AA versus W patients with lung adenocarcinoma. The results propose additional classifiers in the development of clinically meaningful biomarkers.

This reviewer considers the data presented interesting and able to underline the possibility of further stratifications of the tumor classifications that can have important repercussions in the identification
of specific disease paths. Overall it is a good, original research andI suggest its publication also in the present form.

Author Response

Reviewer 2

  • The work presented by Distefano et al. relates specific miRNA isoforms expression patterns that significantly stratify patients in overall survival and relapse-free survival in B / AA versus W patients with lung adenocarcinoma. The results propose additional classifiers in the development of clinically meaningful biomarkers.

This Reviewer considers the data presented interesting and able to underline the possibility of further stratifications of the tumor classifications that can have important repercussions in the identification of specific disease paths. Overall it is a good, original research and I suggest its publication also in the present form.

Response: We appreciate the time and effort that the Reviewer dedicated to providing feedback on our manuscript. Please see attached the revised version of our manuscript. To get our changes in the text quickly, we have labeled the modifications in Red color.